# Nordic Walking Rather Than High Intensity Interval Training Reduced Myostatin Concentration More Effectively in Elderly Subjects and the Range of This Drop Was Modified by Metabolites of Vitamin D

**DOI:** 10.3390/nu13124393

**Published:** 2021-12-08

**Authors:** Katarzyna Micielska, Marta Flis, Jakub Antoni Kortas, Ewa Rodziewicz-Flis, Jędrzej Antosiewicz, Krystian Wochna, Giovanni Lombardi, Ewa Ziemann

**Affiliations:** 1Department of Physical Education and Lifelong Sports, Poznan University of Physical Education, 61-871 Poznan, Poland; micielska.katarzyna@gmail.com; 2Doctoral School, Gdansk University of Physical Education and Sport, 80-336 Gdansk, Poland; 3Department of Physiology, Gdansk University of Physical Education and Sport, 80-336 Gdansk, Poland; kozl.marta@gmail.com; 4Department of Health and Life Sciences, Gdansk University of Physical Education and Sport, 80-336 Gdansk, Poland; jakubantonikortas@gmail.com; 5Department of Physiotherapy, Gdansk University of Physical Education and Sport, 80-336 Gdansk, Poland; ewu.rodziewicz@wp.pl; 6Department of Bioenergetics and Physiology of Exercise, Medical University of Gdansk, 80-210 Gdansk, Poland; jant@gumed.edu.pl; 7Laboratory of Swimming and Water Lifesaving, Poznan University of Physical Education, 61-871 Poznan, Poland; kwochna@awf.poznan.pl; 8Laboratory of Experimental Biochemistry and Molecular Biology, IRCCS Istituto Ortopedico Galeazzi, 20161 Milan, Italy; giovanni.lombardi@grupposandonato.it; 9Department of Athletics, Strength and Conditioning, Poznan University of Physical Education, 61-871 Poznan, Poland

**Keywords:** decorin, aging, myokines, osteokines, 25(OH)D_3_, 24,25(OH)_2_D_3_, 3-epi-25(OH)D_3_

## Abstract

The COVID-19 pandemic and subsequent self-isolation exacerbated the problem of insufficient amounts of physical activity and its consequences. At the same time, this revealed the advantage of vitamin D. Thus, there was a need to verify the effects of those forms of training that can be performed independently. In this study, we examined the effects of Nordic walking (NW) and high intensity interval training (HIIT) with regard to the impact of the metabolite vitamin D. We assigned 32 overweight adults (age = 61 ± 12 years) to one of two training groups: NW = 18 and HIIT = 14. Body composition assessment and blood sample collection were conducted before starting the training programs and a day after their completion. NW training induced a significant decrease in myostatin (*p* = 0.05) concentration; however, the range was dependent on the baseline concentrations of vitamin D metabolites. This drop was accompanied by a significant negative correlation with the decorin concentration. Unexpectedly, NW caused a decrement in both forms of osteocalcin: undercarboxylated (Glu-OC) and carboxylated-type (Gla-OC). The scope of Glu-OC changes was dependent on a baseline concentration of 25(OH)D_2_ (r = −0.60, *p* = 0.01). In contrast, the HIIT protocol did not induce any changes. Overall results revealed that NW diminished the myostatin concentration and that this effect is more pronounced among adults with a sufficient concentration of vitamin D metabolites.

## 1. Introduction

An insufficient level of physical activity, along with a sedentary lifestyle, is particularly common among middle-aged and elderly persons, which in consequence contributes to chronic diseases’ development and to premature mortality [1,2]. With the onset of the COVID-19 pandemic and the related extended periods of self-isolation, this problem was intensified, meaning that two global pandemics coexisted at the same time [3,4]. Paradoxically, Sallis et al. reported that inactivity is a modifiable variable for severe post-COVID-19 outcomes. Among adult patients who had suffered from COVID-19, consistent inactivity (documented in their electronic health record, before the pandemic) represented a significant risk factor for hospitalisation and mortality [5]. Additionally, it has been suggested that a low concentration of 25-hydroxyvitamin D (25(OH)D) contributes to low physical performance, muscle weakness [6] and higher rates of COVID-19 infection and mortality [7,8]. Consequently, the role of these two factors, exercise (especially forms that accessible in periods of lock-down/isolation) and level of vitamin D, is particularly significant.

Nordic walking (NW) is one of the most popular forms of physical activity that can be performed individually. Its beneficial impact is well documented in terms of an improved quality of life and motor skills development [9,10]. One report linked beneficial effects of NW training with the baseline vitamin D concentration of > 20 ng⋅mL^−1^. A reduction of pro-inflammatory markers was more pronounced than among subjects characterized by a lower vitamin D concentration [11]. Alternatively, high intensity interval training (HIIT) was also proposed as a time-saving and advantageous practice to induce adaptive changes [12,13], reduce fall and fracture risk [14] and improve glucose homeostasis and insulin sensitivity among elderly diabetic patients [15]. What is more, HIIT was shown to be an effective way to prevent sarcopenia onset and progression [16].

Regulation of metabolism is associated with proteins released into the bloodstream from different tissues e.g., muscles (myokines), bone cells (osteokines) or adipose tissue (adipokines), which are collectively referred to as exerkines when released in response to physical activity. Exerkines create endocrine-like signalling pathways between distance tissues, by means of which they are considered to mediate adaptation changes to exercise [17]. However, a detailed mechanism of their secretion and action following training is not fully understood. Decorin is one of the myokines that might be released in response to exercise. Although Kanzleiter et al. described a significantly higher decorin expression in response to regular resistance and endurance training among men, as well on an animal model [18], a direct impact of exercise on decorin concentration in women is debated. It was reported that 5 weeks of high intensity circuit training did not alter decorin concentration in middle-aged women [19]. However, decorin was demonstrated to bind with and inhibit myostatin activity, a strong negative regulator of muscle growth [20] and one of the potential serum biomarkers of sarcopenia [21]. Another exerkine associated with muscle function is osteocalcin. The signalling of osteokine in myofibers prevents muscle function deterioration during aging, as well determines muscle adaptation to exercise [22]. Still, the effect of exercise on myostatin and osteocalcin is indistinct. It is worth noting that osteocalcin is under investigation for its potential insulin sensitivity-modulating properties [23,24], especially by its active form, the undercarboxylated osteocalcin (Glu-OC), as it might regulate the insulin sensitivity of adipocytes, just through the secretion of adiponectin [25]. Nonetheless, it was revealed that the beneficial effects of exercise on exerkines can be modulated by vitamin D and possibly by other factors [11].

Since both HIIT and NW can be used interchangeably and can be performed in home/safe-distance conditions, they are considered particularly convenient exercise options in the COVID-19 pandemic. Thus, the aim of this study was to compare the effects of these two training protocols (HIIT vs NW) on myokines’ and osteokines’ secretion, including the impact of some metabolites of vitamin D: 25(OH)D_3_, 25(OH)D_2,_ 24,25(OH)_2_D_3_ and 3-epi-25(OH)D_3_. The study also aimed to assess whether short-term HIIT can be an effective alternative to long-term NW with respect to pro-health changes.

## 2. Materials and Methods

### 2.1. Design of the Study

The study was performed just after the summer holiday (at the beginning of September). One week prior to the start of the experiment, as well as 24 h after the completion of each training protocol, the subjects were tested for body composition analysis and underwent blood collection. The study was approved by the Bioethical Committee of the Regional Medical Society in Gdansk (KB-34/18) in accordance with the Declaration of Helsinki.

### 2.2. Subjects

Thirty-two healthy middle-aged-to-elderly adults (nine men and twenty-three women; age = 61 ± 12 years; body mass = 78 ± 17.5 kg; body mass index, BMI = 27 ± 4.2 kg⋅m^−2^; percent body fat, PBF = 33 ± 7.7%) took part in the study. The participants were characterized by two adiposities to muscle ratios (Table 1) that are good indicators of sarcopenic obesity and physical disability [26]. Moreover, the subjects were characterized by hepatic insulin resistance expressed by a quantitative insulin sensitivity check index, QUICKI = 0.358 ± 0.04, homoeostasis model assessment of insulin resistance, HOMA-IR = 2.02 ± 1.42 and bone mineral content, BMC = 3.0 ± 0.7 kg. The recruitment was encouraged through advertisements in the local and social media. Before the start of the intervention, all participants underwent a medical examination in order to exclude any contraindications to physical activity, such as bone disease, diabetes, uncontrolled hypertension, cardio-respiratory disorders or any orthopaedic issues. Participants included in the study reported that they did not use vitamin D or other supplements; however, some reported using prophylactic medications. Moreover, participants were asked not to change this or other daily habits during the intervention. Subjects were familiarized with the experimental procedures and a written, informed consent was obtained from all of them. Enrolled individuals were assigned to one of two training groups: NW group (*n* = 18, BMI = 26 ± 3.5 kg⋅m^−2^, PBF = 32 ± 7.7%) and HIIT group (*n* = 14, BMI = 29 ± 4.0 kg⋅m^−2^, PBF = 34 ± 8.0%). The NW group completed 12-week training programs, whereas the program for the HIIT group lasted 2 weeks. One week before the start of the experiment, the individual workload for the participants from the HIIT group was determined. Each participant had to perform a preliminary test on the cycle-ergometer by pedalling for 1 min with a cadence of 80–100 rpm, at 90% of the maximum heart rate (HR_max_), with an individually established load (1.5–2.5 W·kg^−1^) to determine the proper workload. Between each bout, a 1 min rest was applied. Participants from the NW group received and were instructed on how to use a sport-tester device (Polar Electro Oy, Professorintie 5, Kempele, Finland M200) for their own cardiovascular and training unit intensity control. Only subjects whose training attendance was at 100% in the HIIT group and at 90% in the NW group were included in the statistical analysis.

### 2.3. Body Composition Assessment

Body mass (BM), body mass index (BMI), bone mineral content (BMC), body fat (BF) and free fat mass (FMM) were evaluated in a fasted state in the morning by using a multi-frequency impedance (1, 5, 50, 250, 500, and 1000 kHz) with the analyser InBody 720 (Biospace, Seoul, Korea). Percent of body fat (PBF) mass repeated measurement precision was expressed as the coefficient of variation, on average, 0.6% [27]

### 2.4. Blood Collection

Blood samples were taken at baseline and 24 h after completing training protocols. Blood collected from the antecubital vein into vacutainer tubes by a professional nurse was centrifuged at 2000× *g* for 10 min at 4 °C and stored at −80 °C until assayed.

The concentrations of myostatin, adiponectin and decorin were determined via ELISA kits, according to the manufacturer’s instructions. For serum myostatin and adiponectin (R&D Systems, Minneapolis, USA catalogue no. DGDF80 and DRP300, respectively) maximal intra-assay coefficient of variability (CV), inter-assay CV and detection sensitivity were 5%, 6%, and 5.32 pg⋅mL^−1^ and 5%, 7%, and 0.891 ng⋅mL^−1^, respectively. Decorin was assessed by Human Decorin DuoSet ELISA (R&D Systems, Minneapolis, USA, catalogue no. DY143, and Ancillary Reagent Kit catalogue no. DY008). Serum undercarboxylated osteocalcin (Glu-OC) and carboxylated-type of osteocalcin (Gla-OC) were assessed by enzyme immunoassay commercial kits Takara Bio Inc., Kusatsu, Japan (catalog no.#MK118 and #MK111, respectively). The detection limits and CV for Glu-OC were 0.25 ng·mL^−1^ and 4.6%, and 0.2 ng·mL^−1^ and 3.3% for Gla-OC.

Insulin was assessed using an immunoassay kit from DiaMetra, Perugia, Italy (catalogue no. DKO076) within intra-assay CV ≤5% and the inter-assay CV ≤10%.

Glucose concentration was assayed on Cobas 6000 (Roche Diagnostics, Warsaw, Poland). In order to define each participant’s insulin sensitivity and insulin resistance, the two well-described indexes were used as follow: the quantitative insulin sensitivity check index (QUICKI; used formula: QUICKI = 1/(log serum insulin μU⋅mL^−1^ + log fasting plasma glucose mg⋅dL^−1^ [28]) and homoeostasis model assessment of insulin resistance (HOMA-IR; used formula: HOMA-IR = fasting serum insulin µU⋅mL^−1^ × fasting plasma glucose mmol⋅L^−1^/22.5 [29]).

Total cholesterol (TC), high-density (HDL), and low-density lipoproteins (LDL) cholesterol, and triglycerides (TG) were assessed with commercially available kits using enzymatic methods (Alpha Diagnostics, Warsaw, Poland).

Vitamin D metabolites: 25(OH)D_3_, 25(OH)D_2_, 24,25(OH)_2_D_3_ and 3-epi-25(OH)D_3_ concentrations were determined and corrected to change in plasma volume, as was previously described by Mieszkowski and co-workers [30]. It was performed by quantitative analysis using liquid chromatography, coupled with tandem mass spectrometry (QTRAP^®^4500 (Sciex, Framingham, MA, USA) coupled with ExionLC HPLC system). The measurement of serum samples was performed in the positive ion mode, using electrospray ionization. The Analyst^®^ software was used to collect raw data and MultiQuant^®^ (Sciex, Framingham, MA, USA) was used to process and quantify it. The reagents used in the procedure were as follows: derivatization agent: 4-(40 -Dimethylaminophenyl)-1,2,4-triazoline-3,5-dione (DAPTAD) synthesized by Masdiag Laboratory (Warsaw, Poland); water; ethyl acetate (POCh S.A., Gliwice, Poland) and methanol (Honeywell, Sigma-Aldrich, Gillingham, Dorset, UK). For the mobile phases acetonitrile (ACN) (Honeywell, Sigma-Aldrich, Gillingham, Dorset, UK), water (POCh S.A., Gliwice, Poland) and formic acid (FA) (Merck KGaA, Darmstadt, Germany) were used. All solvents were of LC-MS grade.

### 2.5. Applied Training Protocol

#### 2.5.1. NW Training Protocol

The applied NW training program included 36 training sessions, 3 times per week for 12 weeks, and was based on previous published protocol [31,32]. Each NW unit was performed as follows: 10-min warm-up, 45–55-min main NW training and 10-min cool-down. Training sessions were conducted by a qualified NW instructor, who demonstrated and taught proper walking with the pools technique and monitored the intensity of the training workload. Each training unit was performed at 60–70% HR_max_ intensity.

#### 2.5.2. HIIT Protocol

HIIT training procedure was based on Little et al. [15] and consisted of 6 supervised training units performed 3 times per week for 2 weeks. The single HIIT duration time was 25 min, including: 3-min warm-up, 10 × 1-min cycling intervals separated by 10 × 1-min of rest and 2-min cool-down at the end of session. The load of warm-up and cool-down was established at 50 W; however, interval workload was the same as individually determined one week prior to the start of the experiment. Cycling intervals were performed at 80–100 rpm⋅min^−1^ with an intensity of 90% HR_max_. While performing HIIT, participants had constant biofeedback from a screen showing their pedalling cadence and HR so that they could keep those indicators maintained. During the 1-min recovery period, adults could rest cycling freely.

### 2.6. Statistical Calculation

Statistical analyses were performed by using a statistics software package (Statistica 13.1 software, TIBCO Software, Palo Alto, USA). The Shapiro–Wilk tests were used to assess the homogeneity of dispersion from normal distribution. The Brown–Forsythe test was used to evaluate the homogeneity of variance. Repeated measures analyses of variances (rANOVA) were calculated. In case of a significant time × group interaction, post hoc tests for unequal sample sizes were performed to identify significantly different results. The variable’s relationships were measured using the Spearman correlation coefficient. The level of significance was set at *p* < 0.05.

## 3. Results

### 3.1. NW Trainining Program

The applied NW sessions did not change sarcopenic obesity indicators (Table 1), nor were glucose homeostasis indicators, the lipid profile and vitamin D metabolites altered (Table 2). Significant changes were observed in cytokines’ concentrations (Figure 1). NW training caused a significant reduction of the serum myostatin concentration (Figure 1A), whereas values in HIIT participants remained unchanged. The range of changes in the NW group was noted among those subjects with a higher baseline level of all vitamin D metabolites (Figure 2). Subjects with 25(OH)D_3_ over 25 ng⋅mL^−1^ had a slightly higher impact of a drop of myostatin (*p* < 0.01; Figure 2A), than those with 24,25(OH)_2_D_3_ over 2.0 ng⋅mL^−1^ (*p* < 0.01; Figure 2B) and 3-epi-25(OH)D_3_ over 1.3 ng⋅mL^−1^ (*p* < 0.01; Figure 2C). Together with the myostatin decrement, the increase of decorin concentration was observed (Figure 1B) and this result was accompanied by the negative correlation between delta changes in the concentrations of these myokines (Figure 3). Unexpectedly, NW training induced a significant drop of both forms of osteocalcin: Glu-OC and Gla-OC (Figure 1C,D, respectively). Among all participants, regardless of the group, the shifts of myostatin modified the range of the osteocalcin decrease, especially for Gla-Oc (*p* = 0.04). Still, it is worth noting that the baseline concentration of 25(OH)D_2_ inversely correlated with the delta osteocalcin change Gla-Oc (r =−0.59, *p* = 0.01, Figure 4).

NW training induced a slight elevation of adiponectin concentration (from 11,528.89 ± 9370.57 to 14,173.56 ± 10,504.19 ng⋅mL^−1^). None of the measured exerkines correlated with glucose metabolism indicators. 

### 3.2. HIIT Program

The HIIT protocol did not induce significant changes in measured factors. Similarly to the NW group, adiposity to muscle ratios (Table 1), glucose metabolism, lipid profile indicators and vitamin D metabolites (Table 2) remained unchanged. Furthermore, concentrations of resting myostatin and decorin did not shift in response to the HIIT program (Figure 1A,B). Both Glu-OC and Gla-OC forms showed an upward trend (Figure 1C,D, respectively); however, these changes were not statistically significant. Serum adiponectin concentration 24 h after the HIIT program’s completion revealed an opposite tendency in comparison to NW training, with the change between groups being statically significant (Figure 5). The concentration of adiponectin declined in response to HIIT (from 8659.79 ± 6600.87 to 7842.79 ± 6069.32 ng⋅mL^−1^), this drop being insignificant. Although there was a significant, negative correlation between adiponectin and triglycerides (r = −0.67, *p* = 0.01) after the intervention, no significant correlations between glucose homeostasis indicators and exerkines were observed. Comparing the effects of NW and HIIT procedures, significant and opposite changes in all measured exerkines concentrations were recorded (Figure 5).

## 4. Discussion

The main finding of the current study is that NW proved to be more effective in inducing changes in blood exerkines’ concentrations in elderly people than HIIT. Thus, our initial assumption that a low-volume and high-intensity HIIT training could be an effective alternative to a high-volume and low-intensity NW training in inducing pro-health changes has not been confirmed. NW training significantly reduced the myostatin concentration in comparison to HIIT training, with this drop correlating inversely with an increase in the decorin concentration. The range of changes of myostatin in the NW group was higher among those subjects who were characterised by a higher concentration of vitamin D metabolites at baseline: 25(OH)D_3_ (over 25 ng⋅mL^−1^), 24,25(OH)_2_D_3_ (over 2.0 ng⋅mL^−1^) and 3-epi-2(OH)D_3_ (over 1.3 ng⋅mL^−1^). By demonstrating how the attenuation of skeletal muscle mass and function can be potentiated in middle-aged and elderly participants, the obtained results are insightful in connection with the COVID-19 pandemic, when access to physical activity may be limited [33]. They also highlight the importance of maintaining the adequate level of vitamin D in every season, where insufficient vitamin D_3_ levels were found to correlate with the severity of COVID-19 and increased rates of hospitalization [34]. Vitamin D is known to have a multitude of non-calcemic actions. This is due in part to the presence of the vitamin D receptor (VDR) in most tissues and cells, including the skeletal muscle, adipose tissue, endocrine pancreas, immune cells, blood vessels, brain and others. While vitamin D deficiency impairs muscle function in both young athletes and elderly people, where it is thought to predispose falls [35]. In the current study, a higher concentration of vitamin D metabolites correlated with the drop of myostatin among those subjects who had trained NW. Although there were no associations between serum markers of vitamin D and decorin, delta changes of myostatin were determined by delta changes of decorin.

Regarding the fact that aging is associated with an elevated level of myostatin and a decline in the concentration of decorin [36], the data obtained here have particular significance. It is worth noting that the impact of diverse forms of exercise on decorin concentration is not well established yet. Previous studies have shown that resistance training did not alter plasma decorin concentration, neither with [37] nor without [19] nutritional intervention in people of different gender and age (men and women, 21–50 years old). Moreover, Kanzleiter et al. indicated an increase in decorin muscle expression observed in healthy, middle-aged men (40–65 years old) in response to a regular, prolonged (12 weeks) combination of resistance and endurance training [18]. Despite the diversity of the decorin interacting network, two main roles emerge as prominent themes in itsfunction: maintenance of cellular structure and outside-in signalling, culminating in anti-tumorigenic effect [38]. Interestingly, it has been suggested that overexpression of decorin causes a significant growth inhibition of breast cancer tumor xenografts in MDA-MB-231 cells [39]. Thus, our data support a growing body of evidence that regular exercise can enhance the circulating decorin concentration; however, more evidence is required to explain its mechanism of beneficial metabolic regulation.

Additionally, Kortas et al. demonstrated that the myostatin concentration among elderly women decreased significantly in response to 12 weeks of NW training, accompanied by a significant increase in osteocalcin and adiponectin concentrations, with serum iron and ferritin determining the effects of training [32]. This study indicated that, in addition to iron, vitamin D can modulate a response to NW training. Circulating myostatin is basically increased in obese subjects; its concentration correlates positively with insulin resistance/pancreatic β-cell dysfunction indicators and negatively with insulin sensitivity indicators [40,41]. The findings of this study do not indicate the amelioration of glucose homeostasis (only slight shifts) and, contrary to data presented by Kortas, the bone-derived osteokines (Glu-OC and Gla-OC) decreased significantly a day after NW training had been completed. It is documented that, in osteoporosis, osteocalcin concentration in elderly women increased in response to 12 weeks of comprehensive physical activity [42]. On the contrary, Wieczorek-Baranowska et al. revealed that 8-week cycle-ergometer training, with a ventilatory threshold intensity at 70–80%, induced a significant decrement of circulating osteocalcin in postmenopausal women [43], which is in line with our study’s findings. Still, the above-mentioned studies did not measure Glu-OC, but rather only the circulating osteocalcin concentration. An elevated Glu-OC concentration was recorded in response to acute exercise [44]. In the current study, NW training induced a decrease in Glu-OC, inversely dependant on the baseline concentration of 25(OH)D_2_. At present, we don’t have a credible explanation for this observation and we therefore recommend further research into this area.

Although circulating concentrations of adiponectin, the anti-inflammatory mediator, did not change significantly in response to NW or HIIT programs, the trends of change between groups were opposite and statistically significant. It has been proven that adiponectin can act as a myokine, as it is also expressed by skeletal muscles during contraction [45]. The obtained results suggest that NW is more effective than HIIT in inducing beneficial shifts of adiponectin.

In our study, contrary to the many well-documented beneficial changes of HIIT protocols [46,47], the desired effect was not observed. A previously published study reported an improvement of glucose homeostasis via an increment of skeletal muscle mitochondrial content, mitofusin 2 protein content and an elevation of the number of glucose transporters 4 (GLUT4) among older diabetic patients (mean age of 63 ± 8 years) in response to a 2-week HIIT protocol [15]. In contract, Shaban et al. did not observe any changes in glucose homeostasis indicators, such as serum insulin and HOMA-IR level, in response to HIIT training among obese subjects with type 2 diabetes [48]. In our study, we did not collect blood immediately after exercise, but only at rest before starting and 24 h after completing exercise protocols. For this reason, we may not have noticed short-term changes in the blood glucose concentration.

Several limitations of the study warrant mentioning. Firstly, both training procedures were different types of exercise and their application was based on previously published data. Still, it should be considered to compare the effect of applying those procedures in the same time period. Secondly, the participants’ diet was only the same on the days of blood collection. For the remainder of the study, their diet was not controlled, and participants only were instructed not to change their daily habits.

To sum up, the obtained results show that the beneficial effects of a diminishing myostatin concentration were induced only by the NW training; however, the range of changes was modified by the concentration of vitamin D metabolites.

## 5. Conclusions

In conclusion, this study is the first to demonstrate that endurance exercise in the form of NW induced a decrease in myostatin and an increase in decorin, with these changes being modulated by the vitamin D status. It also suggests that elderly people can experience more benefits from a high-volume and low-intensity NW than high-intensity and low-volume HIIT.

## Figures and Tables

**Figure 1 nutrients-13-04393-f001:**
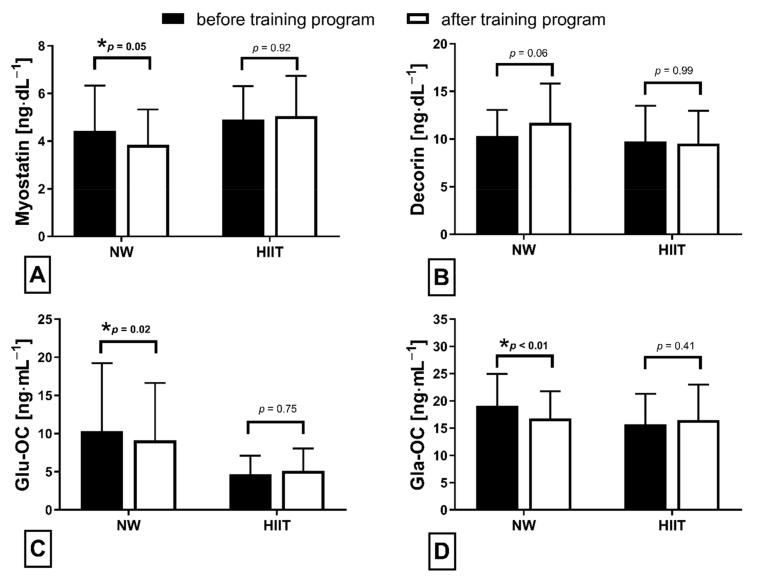
Myokines (**A**,**B**) and osteokines (**C**,**D**) concentration changes in response to applied training protocols: Nordic walking (NW; *n* = 18) and high intensity interval training (HIIT; *n* = 14). Data are presented as mean ± SD; * statistically significant result (post hoc tests); Glu-OC—undercarboxylated osteocalcin; Gla-OC—carboxylated-type of osteocalcin.

**Figure 2 nutrients-13-04393-f002:**
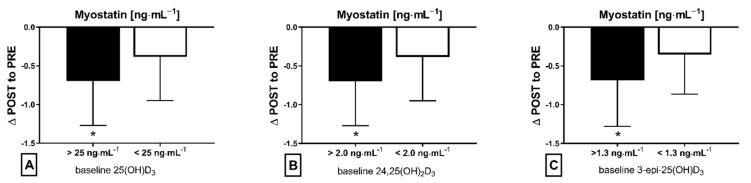
Myostatin concentration delta changes (∆ POST to PRE) dependant on baseline level of metabolite vitamin D in NW training group: (**A**) baseline 25(OH)D_3_, (**B**) baseline 24,25(OH)_2_D_3_ and (**C**) baseline 3-epi-25(OH)D_3_; * statistically significant result (post hoc tests)—*p* < 0.05.

**Figure 3 nutrients-13-04393-f003:**
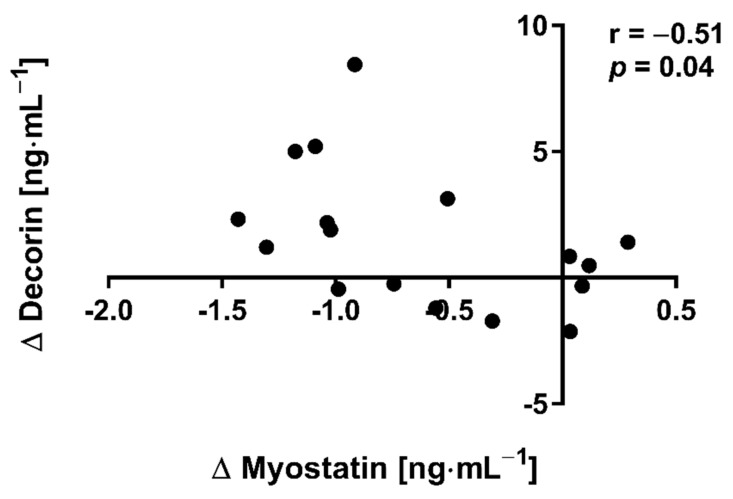
Correlation of myostatin and decorin concentration delta changes (∆ POST to PRE) in response to NW training.

**Figure 4 nutrients-13-04393-f004:**
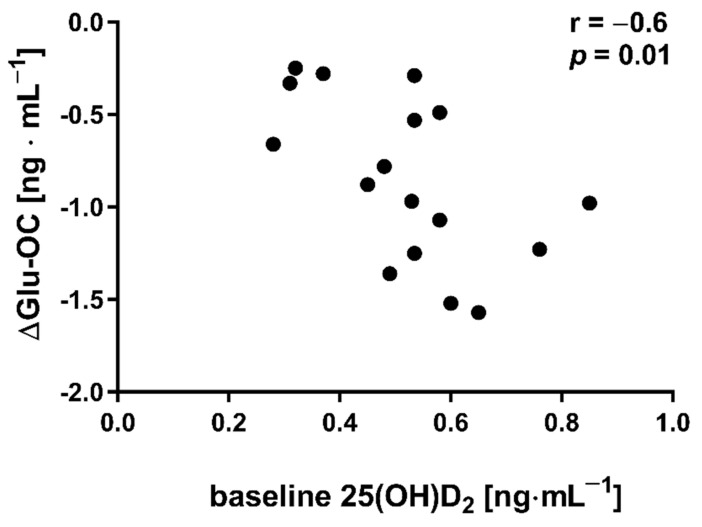
Correlation between baseline concentration of 25(OH)D_2_ and delta changes (∆ POST to PRE) in undercarboxylated osteocalcin (Glu-OC) among participants from NW group.

**Figure 5 nutrients-13-04393-f005:**
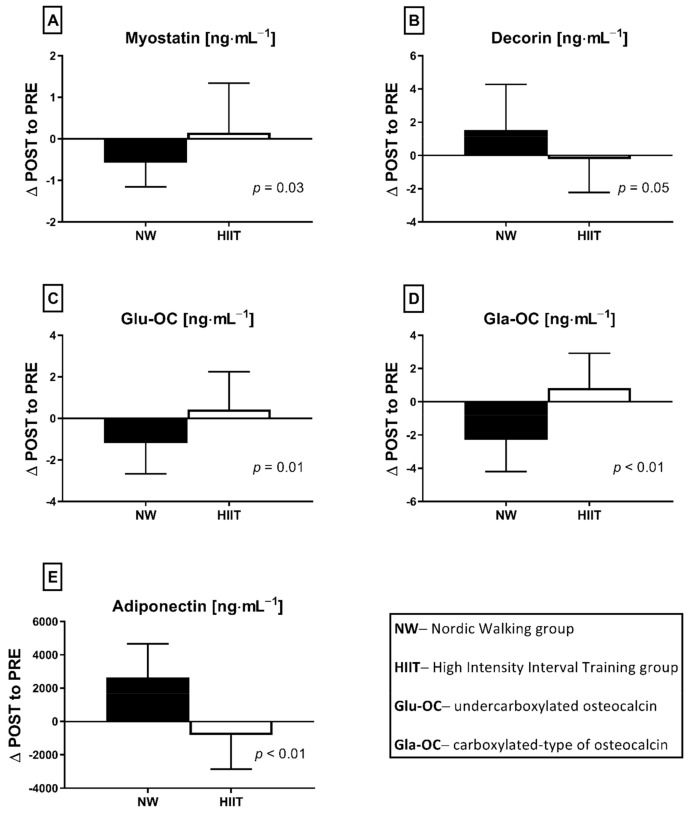
The differences between NW (*n* = 18) and HIIT (*n* = 14) training programs expressed as delta changes (∆ POST to PRE) in myokines (**A**,**B**,**E**) and osteokines (**C**,**D**) concentration, before and after applied interventions. Values are statistically significant. Analysis of variance (rANOVA) was used.

**Table 1 nutrients-13-04393-t001:** Characteristic of participants.

	NW (*n* = 18)		HIIT (*n* =14)		rANOVA
	I	II	I	II	Group × Time Interaction
**Sarcopenic Obesity Indicators**					
BF/FFM [kg⋅kg^−1^]	0.47 + 0.17	0.47 + 0.16	0.51 ± 0.18	0.50 ± 0.19	0.28
BM/FFM [kg⋅kg^−1^]	1.47 + 0.17	1.47 + 0.16	1.51 ± 0.18	1.50 ± 0.19	0.28

Data are presented as mean ± SD; rANOVA- analysis of variance with repeated measure; NW—Nordic walking group, HIIT—high intensity interval training group, I—before the intervention, II—24 h after completing training procedures; BF—body fat, FFM—free fat mass, BM—body mass.

**Table 2 nutrients-13-04393-t002:** Glucose, lipids and metabolite of vitamin D parameters before and after NW and HIIT trainings programs.

	NW (*n* = 18)		HIIT (*n* = 14)		rANOVA
	I	II	I	II	Group × Time Interaction
Glucose homeostasis indicators					
Glucose [mg⋅dL^−1^]	100.83 ± 21.92	93.69 ± 6.01	100.36 ± 8.70	96.29 ± 8.88	0.73
Insulin [µIU⋅mL^−1^]	7.80 ± 4.87	8.19 ± 4.15	7.92 ± 4.90	8.19 ± 6.02	0.62
QUICKI	0.359 ± 0.04	0.357 ± 0.03	0.358 ± 0.04	0.362 ± 0.04	0.42
HOMA-IR	2.02 ± 1.51	1.88 ± 1.10	2.02 ± 1.38	2.01 ± 1.64	0.48
Lipid profile					
Total cholesterol [mg⋅dL^−1^]	232.50 ± 35.05	231.06 ± 35.68	178.93 ± 43.71	176.86 ± 31.34	0.94
HDL cholesterol [mg⋅dL^−1^]	77.75 ± 29.39	73.75 ± 20.43	54.45 ± 15.52	54.79 ± 15.27	0.27
LDL cholesterol [mg⋅dL^−1^]	133.44 ± 37.36	133.5 ± 41.35	104.94 ± 35.48	101.71 ± 28.21	0.63
Triglycerides [mg⋅dL^−1^]	105.63 ± 35.34	118.75 ± 39.74	97.07 ± 61.73	101.43 ± 44.04	0.54
Vitamin D metabolites					
25(OH)D_3_ [ng⋅mL^−1^]	27.61 ± 10.82	27.78 ± 7.86	23.8 ± 5.18	25.54 ± 7.06	0.21
25(OH)D_2_ [ng⋅mL^−1^]	0.52 ± 0.15	0.47 ± 0.11	0.44 ± 0.16	0.45 ± 0.15	0.08
24,25(OH)_2_D_3_ [ng⋅mL^−1^]	2.62 ± 1.59	2.68 ± 1.43	2.12 ± 0.71	2.16 ± 0.83	0.94
3-epi-25(OH)D_3_ [ng⋅mL^−1^]	1.34 ± 0.59	1.61 ± 0.68	1.23 ± 0.41	1.37 ± 0.63	0.27

Data are presented as mean ± SD; rANOVA—analysis of variance with repeated measure; I—before the intervention, II—24 h after completing training procedures; QUICKI—quantitative insulin sensitivity check index; HOMA-IR—homoeostasis model assessment of insulin resistance; HDL cholesterol—high density lipoprotein cholesterol; LDL cholesterol—low density lipoprotein cholesterol, 25(OH)D_3_-25-hydroxyvitamin D_3_, 25(OH)D_2_—25-hydroxyvitamin D_2_, 24,25(OH)_2_D_3_—24,25-dihydroxyvitamin D_3_, 3-epi-25(OH)D_3_—3 epimer of 25-hydroxyvitamin D_3_.

## Data Availability

The data presented in this study are available on request to the corresponding author.

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
