# Peer review of "Nordic Walking Rather Than High Intensity Interval Training Reduced Myostatin Concentration More Effectively in Elderly Subjects and the Range of This Drop Was Modified by Metabolites of Vitamin D"

_nutrients, 2021, doi:10.3390/nu13124393_

Round 1

Reviewer 1 Report

The paper shows an interesting insight but:

line 105: it should be pointed up sex differences

Line 115: do the subjects use drugs or nutritional supplements?

line 120 why a homogeneous sample was not chosen? that is 16/16

line 183-199 why authors choose a so different time of execution? Did they have a training session for both exercises? Why was a bike and not a treadmill was chosen? It is obvious that there is different mechanical load and overall muscle involvement: it is well known that NW is designed to involve non only the lower body but core and arms, bike obviously does not

Protocols should also be optimized better for comparison, such as estimating intensity and Met for each activity

The HIIT 1: 1 protocol is bland also for this reason it may not have provided results

Authors speak of inflammation, why were the interleukins or the TNF not monitored?
Or even inflammation-related miRNAs

Decorin and Myostatin are interesting mediators, but in particular the former still does not have a well-established mechanism and function

Another fundamental difference is that the NW is practiced outdoors the bike indoors, even the sun exposure is therefore different

So I think that the work in the present form cannot be accepted, possibly it could be presented as the analysis of the action of the NW but the comparison makes no sense, so operated

Author Response

Dear Reviewer,

Thank you very much for reviewing our paper and for the opportunity to resubmit it. We have studied your comments carefully and revised the paper accordingly. We hope that you will find that its quality has improved in line with the high standards of the Nutrients.

Reviewer#1 (R1): The paper shows an interesting insight but:

R1: Line 105: it should be pointed up sex differences

Authors (A): We have added this information in the Materials and Methods section. Still, we would like to emphasize that all participating women were post-menopausal, and that the total number of participants was too small to calculate sex-based differences.

R1: Line 115: do the subjects use drugs or nutritional supplements?

A: At baseline, subjects underwent a medical assessment by professional doctor in order to eliminate those with medical contraindications to physical activity. If they reported taking prophylactic medications (such as Polocard or Concor or similar), they were cleared for taking part in the experiment and instructed not to change these habits. We have added this information in the Materials and Methods section.

R1: Line 120: why a homogeneous sample was not chosen? that is 16/16

A: Both training groups initially had equal number of participants. Still, in the statistical analysis, we considered only results from the participants that attended 100% of HIIT and at least 90% of NW training sessions. Due to participants’ absence, eventually the HIIT group counted 14 participants and the NW group 18 participants. We have added this information in the Materials and Methods section.

R1: Line 183-199: why authors choose a so different time of execution? Did they have a training session for both exercises? Why was a bike and not a treadmill was chosen? It is obvious that there is different mechanical load and overall muscle involvement: it is well known that NW is designed to involve non only the lower body but core and arms, bike obviously does not.

A: We had decided to compare two different training protocols that were feasible to perform alone, safely in home conditions. Exercise on a treadmill may be risky and is not recommended to elderly. Therefore, we proposed using a bike, which allows participants to do the training alone, at home. We also had not intended to compare the same training protocol in different conditions (indoor/outdoor). The purpose of the study was to verify if a short-term HIIT protocol could be an alternative to a long-term NW.

We agree with the Reviewer that both these trainings have different patterns of motor units’ involvement. However, previously published research points to an ambiguous effect of NA training on upper limb muscle strength. For example:

  • Ossowski et al (2016) observed no improvement in the handgrip muscle strength in 45 elderly women (63–79 years) with low bone mass (org/10.2147/CIA.S118995).
  • Lee and Park (2015) observed no significant differences in changes in the upper extremity muscle strength among 20 elderly individuals aged ≥70 years. In this study, the upper arm muscle strength was assessed by measuring the number of times a subject was able to lift a 2-kg dumbbell above elbow level during 30 seconds (doi: 1589/jpts.27.2453).
  • Gmiat et al (2017) indeed showed an improvement of muscle strength in the upper limbs in elderly women (DOI: 10.1007/s10522-017-9694-8).

Moreover, since both training programmes could be practiced in home conditions and without supervision, the results of our study can be applied practically in everyday life.

R1: Protocols should also be optimized better for comparison, such as estimating intensity and Met for each activity

A: We agree with the Reviewer that such detailed information would be appropriate; however, we compared the high-intensity, low-volume protocol with the lower-intensity, high-volume protocol.

R1: The HIIT 1: 1 protocol is bland also for this reason it may not have provided results

A: We based the HIIT protocol on a previously published paper from Journal of Applied Physiology (doi:10.1152/japplphysiol.00921.2011), which demonstrated its effectiveness in inducing adaptive changes. This study also was conducted on elderly subjects.

R1: Authors speak of inflammation, why were the interleukins or the TNF not monitored?

Or even inflammation-related miRNAs

A: Based on our previously published paper (doi: 10.1007/s10522-017-9694-8), which reported a reduced concentration of the pro-inflammatory protein— High-Mobility Group Box 1 (HMGB1) and its negative correlation with an elevated irisin concentration in response to the same NW training protocol, we had decided to evaluate HMGB1 concentration also in this study. In particular, because those changes were observed among participants with the baseline vitamin D concentration of more than 20 ng×mL−1.

Still, if  the results of HMGB1 in current study had not a large dispersion in the HIIT group, the differences would probably be significant, because the decrease in the NW group is large. Thus, we decided not to present.

NW (n=18)

HIIT (n=14)

rANOVA

I

II

I

II

Group x time

HMGB1 (pg/ml)

380,77±95,06

330,42±83,55

424,01±179,59

417,8±202,77

0,13

R1: Decorin and Myostatin are interesting mediators, but in particular the former still does not have a well-established mechanism and function

A: We generally agree that the mechanism of decorin is not well-established. Thus, further research is justified. We believe that the results obtained in this study contribute to filling this knowledge gap.

R1: Another fundamental difference is that the NW is practiced outdoors the bike indoors, even the sun exposure is therefore different. So, I think that the work in the present form cannot be accepted, possibly it could be presented as the analysis of the action of the NW but the comparison makes no sense, so operated

A: We do not agree with this statement. It was not the aim of our study to compare effects of indoor/outdoor activity. We assessed the baseline and post-intervention concentrations of vitamin D, and these data clearly showed that there were no changes in vitamin D metabolites’ concentrations in both groups. What is more, the NW intervention was held during autumn/winter period when vitamin D synthesis is null in the geographic location of Poland.

Reviewer 2 Report

The publication by Micielska et al investigated the effects of two separate physical training regiments on the circulating levels of myokines, adipokines, glucose metabolism, vitamin D metabolites, and osteocalcin levels. There reviewer has several moderate concerns about the presentation of the motivation and the data, and the context of the findings presented in the discussion that should be addressed to ensure that the author's data and findings are not lost in a complicated story.

  • The introduction starts with a mention of COVID-19, but the discussion never comes back to this. Why are the authors leading with this? It seems out of place.
  • Similarly, there is a mention of that the data from this studies "support population study demonstrating that regular exercise reduces cancer risk..." I realize that the authors are trying to suggest that this reduction in cancer risk could be due to increased decorin, which has been reported to have anti-cancer properties. However, as stated, the conclusion is too far and I would recommend toning this down as cancer was not an outcome in this study.
  • Overall the motivation for this study in the introduction is pretty convoluted. Paragraph 3 in particular seems to throw a lot of facts out, but it isn't always clear how they are connected. 
  • The motivation of the study is also a bit confusing. As it reads now, it seems like the authors wanted to assess the impact of myokine secretion on glucose and osteokines. If this is indeed stated as intended, why was there no correlations run between myokine change and glucose measures? 
  • When is the baseline blood draw and measurements? How much before training was this measurement made? Did it vary for each of two training regimens? 
  • Similarly, when was the post-exercise blood draw made. The discussion suggests that it was after rest, but I didn't see how many hours, days or weeks past the last training cycle.
  • Were both training plans performed outside? Presumably this would influence vitamin D production, correct? 
  • The statistical analysis doesn't describe what tests were used for the data presented in Figures 2 & 5. I assume that this was a T-test of some kind?
  • Lines 227 to 228 reads "Any significant shifts in adiponectin concentration were recorded." Does this mean that there were no significant changes? Why isn't adiponectin baseline and follow-up measurements presented? 
  • In the figures, I would suggest that the authors present their p-values as p<0.01 rather than p=0.00. 
  • I am not clear that I understand the motivation for presenting the correlation analyses in Figures 3 & 4. Why were these run? Were there a variety of other correlations run that were not significant that are not being shown or mentioned? If the goal was to assess the impact of myokine change on glucose variables, what is the motivation for Figure 4? 
  • The discussion introduces the term "exerkines" for the first time. Does this refer to all of the measured myokines, osteokines, adipokines, etc? 

Author Response

Dear Reviewer,

Thank you very much for reviewing our paper, and for the opportunity to resubmit it. We have studied your comments carefully and revised the paper accordingly. We hope that you will find that its quality has improved in line with the high standards of the Nutrients.

Reviewer#2 (R2): The publication by Micielska et al investigated the effects of two separate physical training regiments on the circulating levels of myokines, adipokines, glucose metabolism, vitamin D metabolites, and osteocalcin levels. There reviewer has several moderate concerns about the presentation of the motivation and the data, and the context of the findings presented in the discussion that should be addressed to ensure that the author's data and findings are not lost in a complicated story.

R2: The introduction starts with a mention of COVID-19, but the discussion never comes back to this. Why are the authors leading with this? It seems out of place.

Authors (A): Following the Reviewer’s suggestion, we have added more literature references to the Discussion section (doi:10.1002/jcsm.12589, doi:10.1080/10408398.2020.1841090) in connection with the practical applicability of the study results and COVID-19-related research.

R2: Similarly, there is a mention of that the data from this studies "support population study demonstrating that regular exercise reduces cancer risk..." I realize that the authors are trying to suggest that this reduction in cancer risk could be due to increased decorin, which has been reported to have anti-cancer properties. However, as stated, the conclusion is too far and I would recommend toning this down as cancer was not an outcome in this study.

A: Following the Reviewer’s suggestion, we have revised the text, especially where it read speculative and could have been misleading.

R2: Overall the motivation for this study in the introduction is pretty convoluted. Paragraph 3 in particular seems to throw a lot of facts out, but it isn't always clear how they are connected. 

A: Following the Reviewer’s suggestion, we have re-written this paragraph to make it clearer and more understandable.

R2: The motivation of the study is also a bit confusing. As it reads now, it seems like the authors wanted to assess the impact of myokine secretion on glucose and osteokines. If this is indeed stated as intended, why was there no correlations run between myokine change and glucose measures? 

A: Following the Reviewer’s suggestion, we have re-written the aim of the study to make it clearer. We have added the information in the Results section that no correlations between exerkines and glucose homeostasis indicators were observed.

R2: When is the baseline blood draw and measurements? How much before training was this measurement made? Did it vary for each of two training regimens? 

A: The baseline blood collection was assessed one week prior to the start of both training protocols. All details are described in the Materials and Methods section.

R2: Similarly, when was the post-exercise blood draw made. The discussion suggests that it was after rest, but I didn't see how many hours, days or weeks past the last training cycle.

A: The post-exercise blood collection was carried out 24 hours after the last training unit. We have now added this information in the text.

R2: Were both training plans performed outside? Presumably this would influence vitamin D production, correct? 

A: HIIT training was performed indoors, whereas NW – outdoors. With that, we assessed the baseline and post-intervention concentrations of vitamin D; however, no changes were recorded in both groups. Thus, the effect of exercise did not depend on the place, where training had been performed.

What is more, the intervention was held during autumn/winter period when vitamin D synthesis is null in the geographic location of Poland (49–54°N; Central Europe). Although the experiment was conducted in the post-summertime period, measured values of vitamin D of our participants were below the recommended level.

A longitudinal study conducted in central Europe reported an increase of vitamin D concentration in summertime (August) reaching 42 ng×mL-1 for children, but only 21 ng×mL-1 for elderly (aged 80–89 years) (doi: 10.1155/2014/589587). This may explain why we recorded below the recommended vitamin D status in our participants - middle-aged-to-elderly adults (age = 61 ± 12 years), regardless of the place where training had been performed (indoors/outdoors).

R2: The statistical analysis doesn't describe what tests were used for the data presented in Figures 2 & 5. I assume that this was a T-test of some kind?

A: Thank you for spotting this. We have added a detailed account of the analyses performed in the figures’ descriptions.

R2: Lines 227 to 228 reads "Any significant shifts in adiponectin concentration were recorded." Does this mean that there were no significant changes? Why isn't adiponectin baseline and follow-up measurements presented? 

A: We have added more information in the Results section and clarified which changes were statistically significant.

R2: In the figures, I would suggest that the authors present their p-values as p<0.01 rather than p=0.00. 

A: Following the Reviewer’s suggestion, we have changed figures p-values and introduced similar changes in the text.

R2: I am not clear that I understand the motivation for presenting the correlation analyses in Figures 3 & 4. Why were these run? Were there a variety of other correlations run that were not significant that are not being shown or mentioned? If the goal was to assess the impact of myokine change on glucose variables, what is the motivation for Figure 4? 

A: In this study, not all baseline assumptions were confirmed upon completion. We could not explain all recorded changes but we consider them interesting for the readers. We have added this statement in the Discussion section.

R2: The discussion introduces the term "exerkines" for the first time. Does this refer to all of the measured myokines, osteokines, adipokines, etc? 

A: Yes, the term “exerkines” refers to the measured myokines, osteokines and adipokines. We mentioned this in the Introduction section, but following the Reviewer’s suggestion, we have re-written the 3rd paragraph to be clearer and more understandable for the readers.

Round 2

Reviewer 1 Report

The authors revised the manuscript reasonably well. For me, now, in this form is suitable for publication.

Reviewer 2 Report

Thanks to the authors for their detailed response to the reviewer comments. The manuscript is more clear. I have only one minor concern and that is in the use of the term prophylactic medications on line 138. Were these prophylactics to prevent metabolic complications or some other condition?